# Ibrexafungerp: A First-in-Class Oral Triterpenoid Glucan Synthase Inhibitor

**DOI:** 10.3390/jof7030163

**Published:** 2021-02-25

**Authors:** Sabelle Jallow, Nelesh P. Govender

**Affiliations:** 1Centre for Healthcare-Associated Infections, Antimicrobial Resistance and Mycoses (CHARM), National Institute for Communicable Diseases, a Division of the National Health Laboratory Service, Johannesburg 2131, South Africa; neleshg@nicd.ac.za; 2School of Pathology, Faculty of Health Sciences, University of the Witwatersrand, Johannesburg 2193, South Africa

**Keywords:** ibrexafungerp (IBX), SCY-078, MK-3118, fungal cell wall, glucan synthase inhibitor, triterpenoid antifungal, fungerp, β-(1,3)-D-glucan, candidiasis, aspergillosis, invasive fungal disease

## Abstract

Ibrexafungerp (formerly SCY-078 or MK-3118) is a first-in-class triterpenoid antifungal or “fungerp” that inhibits biosynthesis of β-(1,3)-D-glucan in the fungal cell wall, a mechanism of action similar to that of echinocandins. Distinguishing characteristics of ibrexafungerp include oral bioavailability, a favourable safety profile, few drug–drug interactions, good tissue penetration, increased activity at low pH and activity against multi-drug resistant isolates including *C. auris* and *C. glabrata.* In vitro data has demonstrated broad and potent activity against *Candida* and *Aspergillus* species. Importantly, ibrexafungerp also has potent activity against azole-resistant isolates, including biofilm-forming *Candida* spp., and echinocandin-resistant isolates. It also has activity against the asci form of *Pneumocystis* spp., and other pathogenic fungi including some non-*Candida* yeasts and non-*Aspergillus* moulds. In vivo data have shown IBX to be effective for treatment of candidiasis and aspergillosis. Ibrexafungerp is effective for the treatment of acute vulvovaginal candidiasis in completed phase 3 clinical trials.

## 1. Introduction

Antifungals that inhibit the biosynthesis of β-(1,3)-D-glucan, an important cell wall component of most fungi, the potential to exhibit potent broad-spectrum of activity [1,2]. These drugs target an enzyme, β-(1,3)-D-glucan synthase that is unique to lower eukaryotes, limiting their toxicity in humans [1,3]. The echinocandins were the first glucan synthase inhibitors approved for use in 2001 [4] and have broad-spectrum activity against most common fungal pathogens (*Candida* spp., *Aspergillus* spp.), except for *Cryptococcus neoformans* [5]. Despite their good efficacy in the treatment of invasive *Candida* infections and low toxicity, their use is limited to parenteral administration only [2,3]. Echinocandins have very high molecular masses of about 1200 kDa [2,6], possibly resulting in their poor oral absorption [3,7,8]. Furthermore, distribution of the first-generation echinocandins to the central nervous system, intraocular fluids, and urine is poor, mainly due to their high protein-binding capabilities (>99%) and high molecular masses [3,7,8]. Active research into new drugs by high throughput screening of natural products from endophytic fungi led to the discovery of enfumafungin, a triterpene glycoside [9]. Enfumafungin is structurally distinct from echinocandins (Figure 1) [10,11], forming a new class of antifungals called “fungerps” (Antifungal Triterpenoid) [12,13,14]. Modifications of enfumafungin for improved oral bioavailability and pharmacokinetic properties led to the development of the semi-synthetic derivative, which was named ibrexafungerp (IBX) [15] by the World Health Organization’s international non-proprietary name group [16].

## 2. Mechanism of Action and Resistance

Ibrexafungerp (formerly SCY-078 or MK-3118) is a first-in-class triterpenoid antifungal that inhibits biosynthesis of β-(1,3)-D-glucan in the fungal cell wall. Glucan represents 50–60% of the fungal cell wall dry weight [17]. β-(1,3)-D-glucan is the most important component of the fungal wall, as many structures are covalently linked to it [17]; furthermore, it is the most abundant molecule in many fungi (65–90%) [17,18], making it an important antifungal target [1,12]. Inhibition of β-(1,3)-D-glucan biosynthesis compromises the fungal cell wall by making it highly permeable, disrupting osmotic pressure, which can lead to cell lysis [19,20,21]. β-(1,3)-D-glucan synthase is a transmembrane glycosyltransferase enzyme complex comprised of a catalytic Fks1p subunit encoded by the homologous genes *FKS1* and *FKS2* [22] and a third gene, *FKS3* [23]; a rho GTPase regulatory subunit encoded by the R*ho1p* gene [24]. The catalytic unit binds UDP-glucose and the regulatory subunit binds GTP to catalyse the polymerization of UDP-glucose to β-(1,3)-D-glucan [25], which is incorporated into the fungal cell wall, where it functions mainly to maintain the structural integrity of the cell wall [19,20,21].

Ibrexafungerp (IBX) has a similar mechanism of action to the echinocandins [26,27] and acts by non-competitively inhibiting the β-(1,3) D-glucan synthase enzyme [12,27]. As with echinocandins, IBX has a fungicidal effect on *Candida* spp. [28] and a fungistatic effect on *Aspergillus* spp. [29,30]. However, the ibrexafungerp and echinocandin-binding sites on the enzyme are not the same, but partially overlap resulting in very limited cross-resistance between echinocandin- and ibrexafungerp-resistant strains [26,27,31]. Resistance to echinocandins is due to mutations in the *FKS* genes, encoding for the catalytic site of the β-(1,3) D-glucan synthase enzyme complex; specifically, mutations in two areas designated as hot spots 1 and 2 [32,33], have been associated with reduced susceptibility to echinocandins [33,34]. The β-(1,3) D-glucan synthase enzyme complex is critical for fungal cell wall activity; alterations of the catalytic core are associated with a decrease in the enzymatic reaction rate, causing slower β-(1,3) D-glucan biosynthesis [35]. Widespread use and prolonged courses of echinocandins have led to echinocandin resistance in *Candida* spp., especially *C. glabrata* and *C. auris* [36,37,38,39,40]. Ibrexafungerp has potent activity against echinocandin-resistant (ER) *C. glabrata* with *FKS* mutations [41], although certain *FKS* mutants have increased IBX MIC values, leading to 1.6–16-fold decreases in IBX susceptibility, compared to the wild-type strains [31]. Deletion mutations in the *FKS1* (F625del) and *FKS2* genes (F659del) lead to 40-fold and >121-fold increases in the MIC_50_ for IBX, respectively [31]. Furthermore, two additional mutations, W715L and A1390D, outside the hotspot 2 region in the *FKS2* gene, resulted in 29-fold and 20-fold increases in the MIC_50_ for IBX, respectively [31]. The majority of resistance mutations to IBX in *C. glabrata* are located in the *FKS2* gene [31,40], consistent with the hypothesis that biosynthesis of β-(1,3) D-glucan in *C. glabrata* is mostly mediated through the *FKS2* gene [32].

## 3. Important Pathogenic Fungi and Antifungal Spectrum

Invasive fungal infections (IFIs) are usually opportunistic [42]. The incidence of IFIs has been increasing globally due to a rise in immunocompromised populations, such as transplant recipients receiving immunosuppressive drugs; cancer patients on chemotherapy, people living with HIV/AIDS with low CD4 T-cell counts; patients undergoing major surgery and premature infants [42,43]. IFIs are a major cause of global mortality with approximately 1.5 million deaths per annum [44]; mainly due to *Candida*, *Aspergillus, Pneumocystis*, and *Cryptococcus* species [44]. Furthermore, there is an increase in antifungal resistance limiting available treatment options [45,46]; a shift in species causing invasive disease [47,48,49,50] to those that may be intrinsically resistant to some antifungals [51,52]. Several fungal pathogens (e.g., *Candida auris, Histoplasma capsulatum, Cryptococcus* spp., *Emergomyces* spp.) are gaining importance, especially in middle-income countries such as South Africa, India, Brazil and Colombia.

*Candida auris* has been reported in over 39 countries as an important emerging fungal pathogen [48] with a high crude mortality rate and a propensity for multidrug resistance [53,54,55,56,57,58,59]. *C. auris* has also been reported as an important cause of nosocomial outbreaks [60,61] due to its ability to colonize skin, form biofilms and resist standard disinfectants; due to its ease of person-to-person and person-to-environment transmission [60,61,62]. Within the last decade, *C. auris* became the third most common cause of candidaemia in South Africa, causing >10% of all culture-confirmed cases of invasive candidiasis [49,63,64]. A large proportion of *C. auris* infections are fatal due to the comorbidities in these patients, but multidrug- or even pan-resistance to available antifungals may also contribute to inappropriate treatment and adverse outcomes [53,54,55,56,57,58,59].

Invasive aspergillosis, a life-threatening acute disease, has a reported mortality of up to 85% [65,66]. First-line treatment is with voriconazole, though resistance to the azole class of drugs has been reported [67,68,69]. Resistance to amphotericin B formulations, used as alternative therapy, is rarer, although *Aspergillus terreus* is intrinsically amphotericin B-resistant [47]. 

*Pneumocystis* infections [70] have gained importance in the human immunodeficiency virus (HIV) era as an acquired immunodeficiency syndrome (AIDS)-related opportunistic infection [70]. *Pneumocystis jirovecii* causes *Pneumocystis* pneumonia in humans (PCP), which is still a leading cause of opportunistic infection in HIV/AIDS patients, even in the era of combination antiretroviral therapy [71]. In addition, *Pneumocystis* infections are increasing in non HIV-infected populations with impaired cell–mediated immunity, including those on immunomodulatory drugs or with underlying medical conditions such as inflammatory or autoimmune diseases [72,73]. First-line treatment is usually with trimethoprim-sulfamethoxazole (TMP-SMX) (alternatives include clindamycin-primaquine, atovaquone and pentamidine) [74] instead of the known antifungal classes. *Pneumocystis jirovecii* utilizes cholesterol, a mammalian-associated sterol, instead of ergosterol [75]; whose biosynthetic pathway is exploited by most antifungals, leading to intrinsic resistance of *Pneumocystis* spp. to these drugs [74]. Resistance mutations in the dihydropteroate synthase and cytochrome *bc1* genes against TMP-SMX and atovaquone have already been identified [76]. *Pneumocystis* spp. are sensitive to glucan synthase inhibitors; however, owing to their unique life cycle, only the ascus (cyst) forms and not the trophic forms are sensitive to these drugs [74]. Glucan synthase inhibitors can therefore, only control, but not eradicate PCP colonization/infection [74].

The activity of glucan synthase inhibitors depends on the proportion of β-(1,3) D-glucan in the fungal cell wall, which can differ in different fungal species [77]. Most *Saccharomyces, Candida* and *Aspergillus* species, are susceptible to glucan synthase inhibitors [7,20,26,41,78,79,80], because β-(1,3) D-glucan is dominant in their cell walls [77]. These drugs also have activity against the ascus form of *Pneumocystis jirovecii* [81]. Fungi, such as those in the order Mucorales, *Fusarium* spp. and *Scedosporium* spp. with limited or no β-(1,3) D-glucan, are intrinsically resistant to this class of drugs [82]. However, a paradox occurs in *Cryptococcus neoformans* whose cell wall contains predominantly β-(1,3) D-glucan, yet is tolerant to this class of drugs [5,83].

## 4. In Vitro Activity

***Yeasts:*** In vitro analysis of ibrexafungerp showed potent activity against a broad spectrum of >1300 *Candida* isolates [41,80,84,85,86,87,88,89,90,91]. Activity against *C. lusitaniae* and *C. krusei* (MIC range: 1–2 and 0.5–4 µg/mL, respectively), seemed to be less potent compared to other *Candida* spp. [87]. Compared to echinocandins, IBX was generally less potent (higher MIC values) for most common *Candida* spp., except for *C. parapsilosis* [86,87]. Ibrexafungerp showed strong activity against azole-resistant isolates (including *C. albicans, C. parapsilosis, C. tropicalis, C. auris, C. krusei, C. glabrata, C. guilliermondii, C. lusitaniae, C. inconspicua*) [84,87,90]; however, activity against echinocandin-resistant *FKS* mutants (*C. albicans, C. krusei, C. tropicalis, C. glabrata, C. auris*) was variable [84,86]. IBX has more activity against a majority of *FKS* mutants compared to the echinocandins, with 70–86% of echinocandin-resistant mutants susceptible to IBX compared to 17–50% for the echinocandins [80,84,86,87,88,89], possibly because selection of these mutants were due to echinocandin exposure. Most *Candida* echinocandin-resistant *FKS* mutants were susceptible to IBX [26,84,87,88], especially *C. glabrata* [41,87,89] and *C. auris* isolates [57,90], but some mutants with the F641S, F649del, F658del and F659del mutations had reduced susceptibility to IBX [80,84,86,89]. Ibrexafungerp produced enhanced activity against echinocandin-resistant *C. albicans* and *C. glabrata,* compared to caspofungin [80]. IBX also demonstrated potent activity against pan-resistant (resistance to ≥2 azoles, all echinocandins and amphotericin B) *C. auris* isolates from a New York City outbreak [57]. *Candida* biofilms use multiple resistance mechanism to escape from antifungals; leading to inherent resistance to azoles [92,93]. β-(1,3)-D-glucan is a key component of biofilm constituent; thus, it is not surprising that IBX has shown activity against different biofilm-producing *Candida* species (*C. albicans, C. parapsilosis, C. tropicalis, C. glabrata*) [41,84].

Among 13 other non-*Candida* yeasts including *Rhodotorula mucilaginosa, Trichosporon* spp. (*T. asahii, T. dermatis, T. inkin, T. japonicum*) and *Arxula adeninivorans,* IBX activity was variable ranging from 0.25–≥128 µg/mL [84]. In another study of 100 non-*Candida* yeasts, IBX showed activity against *Malassezia pachydermatis* (MIC: 0.5 μg/mL), *Pichia* spp. (MIC: 0.5–1 μg/mL) and to some *Trichosporon mucoides* (MIC range: 0.125–2 μg/mL) [94]. In vitro analysis of IBX, at different pH levels against *Candida* spp., showed increased potency at lower pH, with MIC_90_ values of 0.5, 0.25 and <0.016 μg/mL at pH levels of 7.0, 5.72 and 4.5, respectively; indicating increased activity at low pH (owing to its pH-dependent solubility) that may increase potency for treatment of vulvovaginitis [95].

***Moulds:*** Glucan synthase inhibitors have a fungistatic effect on some moulds [29,30], such as *Aspergillus* spp., despite high proportion of β-(1,3) D-glucan in the *Aspergillus* cell wall [29]. While these drugs may not kill these species of mould, they can have a profound antifungal effect both in vitro and in vivo [29,30]. Treatment with glucan synthase inhibitors causes lysis of growing hyphal tips; leading to short, stubby, highly branched hyphae or abnormal hyphal growth [29,96]. The fungistatic effect of these drugs means that MIC values are not accurate endpoint measures; instead minimum effective concentrations (MEC) (the lowest concentration at which abnormal hyphal growth occurs) are usually used in antifungal susceptibility testing [96]. IBX showed potent in vitro inhibition of *Aspergillus* species complexes (*A. fumigatus, A. niger, A. flavus, A. terreus, A. nidulans, A. glaucus, A. ustus, A. versicolor, A. westerdijkiae, A. tamarii, A. calidoustus*), including azole-resistant strains [26,30,80,94]. Echinocandin-resistant (ER) *Aspergillus* spp. are very rare, although an *A. fumigatus* mutant (S678P) has been described [97]. IBX showed increased activity against the S678P ER mutant with a MIC value that was 133-fold less than that observed for caspofungin [26]. 

IBX activity against medically important non-*Aspergillus* moulds including the Order Mucorales (*Rhizopus, Mucor, Rhizomucor, Cunninghamella, Lichtheimia* species), *Fusarium* spp., *Scedosporium* spp., *Paecilomyces* spp., and *Scopulariopsis* spp. showed variable results [98]. IBX was very active against *Paecilomyces variotii, Penicillium citrinum, Scytalidium dimidiatum, Alternaria* spp. *and Cladosporium* spp.; but had limited to no activity against the Mucorales, *Fusarium* spp., *Purpureocillium lilacinum, Lichtheimia coerulea, Lichtheimia corymbifera, Acremonium* spp., *Cladosporium cladosporioides, Trichoderma citrinoviride* and *Trichoderma longibrachiatum* [94,98]. IBX showed variable activity against *Scopulariopsis* spp. and modest activity against *Scedosporium apiospermum* and *Lomentospora* (formerly *Scedosporium*) *prolificans* [98]. Interestingly, IBX was the only drug, amongst those tested, that had any activity against the pan-resistant *Lomentospora prolificans* isolates [98].

**Other fungi:** Among dermatophytes, IBX demonstrated potent activity against Microsporum canis, Trichophyton tonsurans, Trichophyton mentagrophytes, and Trichophyton rubrum [94].

## 5. In Vivo Data from Animal Models

In a neutropenic murine model of invasive candidiasis, ibrexafungerp administered orally every 12 h showed in vivo activity against both wild type (30 mg/kg) and echinocandin-resistant (40 mg/kg) *C. glabrata* strains, while caspofungin showed activity against the wild type strains only [99]. IBX given orally showed activity against *C. albicans, C. glabrata,* and *C. parapsilosis* in a neutropenic murine model of disseminated candidiasis, [100,101]. In an in vivo study of *C. auris* skin colonization in guinea pigs, 10 mg/kg oral dose of IBX produced a significantly lower fungal burden in the treated guinea pigs compared to those untreated; however with higher IBX doses (20 and 30 mg/kg) and with micafungin (5 mg/kg), the reduction in fungal burden was not significant [102]. At the end of treatment, histopathology results showed no fungal elements in all treated animals, while the untreated animals had fungal elements present, indicating that all treatment arms were able to clear the *C. auris* colonization [102]. Intra-abdominal candidiasis (IAC) is a difficult-to-treat invasive disease due to poor drug penetration at the site of infection and hence associated with high mortality [103,104]. In a mouse model of IAC, ibrexafungerp exhibited good penetration with robust accumulation within intra-abdominal lesions [103]. IBX concentrations in liver abscesses were 100-fold higher compared to that in serum [104].

IBX also demonstrated potent activity against both wild-type and azole-resistant strains of *A. fumigatus* in a murine model of invasive aspergillosis; with significant increase in survival and corresponding reductions in fungal kidney burden and serum galactomannan levels in treated mice compared to untreated mice [105]. Intravenous IBX (7.5 mg/kg/day) in combination with oral isavuconazole (40 mg/kg/day) showed potent activity in a neutropenic model of rabbit invasive pulmonary aspergillosis [106]. This drug combination demonstrated increased survival, reduced fungal pathogen-mediated pulmonary injury, decreased galactomannan antigenemia and serum (1,3)-β-D-glucan levels compared to either drug alone [106].

A prophylactic murine model of *Pneumocystis* pneumonia (PCP) found that oral IBX (30 mg/kg BID) reduced total nuclei and asci counts in lung tissue and improved survival; similar results were obtained with TMP/SMX, the gold standard for PCP therapy [81]. IBX reduced number of asci significantly by day 7 with asci being microscopically undetectable by day 14, in a therapeutic murine model of *Pneumocystis* pneumonia [107]. However, compared to TMP/SMX, total nuclei was only reduced, but was still detectable in the IBX group; survival was better for the TMP/SMX group [107].

## 6. Clinical Efficacy

Currently, there are 12 listed clinical trials for ibrexafungerp (Table 1), eight of which have been completed (https://ClinicalTrials.gov/; accessed on 8 January 2021). Clinical data from at least 1000 participants using both single and multiple daily doses of IBX, as high as 1600 mg, revealed a safe and tolerable profile [108,109,110,111]. Mild adverse events were reported including headaches and gastrointestinal issues such as, diarrhea, nausea, vomiting and abdominal discomfort [108,109,110,111].

In a prospective phase 2 (NCT02244606) randomized, open-label, multi-centre study in patients with invasive candidiasis including candidaemia, IBX administered as an oral step-down treatment after echinocandin therapy, was compared to the standard of care (SOC) treatments: either oral fluconazole or intravenous micafungin for fluconazole-resistant isolates [108]. Efficacy was determined by assessment of global response, with a favourable global response defined as resolution of signs and symptoms (clinical response) and negative *Candida* cultures (microbiological response), evaluated at the end of therapy [108]. The global response was similar between the IBX (500 mg: 71%; 750 mg: 86%) and SOC (75%) arms, although 750 mg of IBX gave a higher response rate [108]. An ongoing, open-label, single-arm, phase 3 study (CARES: NCT03363841), expected to end in May 2021, is evaluating the efficacy of IBX in patients with *Candida auris* infections. In preliminary results, infection were completely resolved (culture negative) in two cases after treatment with IBX, including a case with difficult-to-treat *C. auris* that persisted after treatment with fluconazole and micafungun [112].

A phase 2, randomized, double-blind, double-dummy, dose-finding study (DOVE: NCT03253094) was done to compare the efficacy of oral ibrexafungerp to oral fluconazole (FLU) in patients with acute vulvovaginal candidiasis (VVC) [113]. The primary endpoints were clinical cure (complete resolution of all signs and symptoms) and mycological eradication (negative culture for yeast) at the test of cure (TOC) visit on day 10 [113]. The clinical cure (52% vs. 58%) and mycological eradication (63% vs. 63%) rates were similar for IBX and FLU, respectively; however, after 25 days, clinical cure (70% vs. 50%) and mycological eradication (48% vs. 38%) rates were higher for IBX compared to FLU, respectively [113]. In two phase 3 randomized, double-blind, placebo-controlled clinical trials in patients with acute vulvovaginal candidiasis, VANISH 303 (NCT03734991) and VANISH 306 (NCT03987620), with the same end points as the DOVE study, complete resolution of all vaginal signs and symptoms by test of cure (day 10) date was significantly higher in the IBX groups compared to placebo [114,115]. In VANISH 303, clinical cure, mycological eradication, clinical improvement at TOC date and complete resolution of symptoms at day 25 were 51% vs. 29%, 50% vs. 19%, 64% vs. 37%, and 60% vs. 45%, respectively, in the IBX group compared to the placebo [114]. Similarly in VANISH 306, clinical cure, mycological eradication, clinical improvement and resolution of symptoms were 63% vs. 44%, 59% vs. 30%, 72% vs. 55%, and 74% vs. 52%, respectively, in the IBX group compared to the placebo [115]. A large (320 participants) multicentre, randomized, double-blind phase 3 study (CANDLE: NCT04029116) to investigate the efficacy of IBX compared placebo in participants with recurrent vulvovaginal candidiasis is currently ongoing and expected to end in September 2021 [116].

A phase 3 open-label single-arm study (FURI: NCT03059992) is investigating the efficacy of IBX in patients, with *Candida* and *Aspergillus* disease, who are either intolerant of or refractory to standard of care antifungal treatment [117]. The primary end-point is global success, defined as composite assessment of clinical, microbiological, serological and/or radiological responses at end of treatment [117]. Preliminary data have shown that the majority of the patients (83%) had either a complete or partial response (56%) or stable disease (27%), but 15% of the patients did not respond to IBX and 2% were indeterminate [117]. The FURI study was expanded to include other fungal diseases such as coccidioidomycosis, histoplasmosis, blastomycosis and other emerging fungi (Table 1).

## 7. Pharmacokinetics/Pharmacodynamics

Ibrexafungerp is the first orally-available glucan synthase inhibitor, with about 50% bioavailability in animal models [110]. In vitro studies using caco-2 cell monolayers and 5μM of IBX gave an average permeability of 8.9 ± ×10^−6^ cm/s in the apical-to-basolateral direction, indicating good oral absorption [110]. In vivo efficacy data from 3 murine models of disseminated candidiasis produced potent activity against *C. albicans* after 7 days of twice-daily oral treatment with IBX; with a mean therapeutic exposure of 14.3 μM·h across all models [110]. In other animal models, IBX was well absorbed into plasma after oral dosing with bioavailability values of >51, 45, and 35%; half-lives (t_1/2_) of ~8.3, 9.1, and 15.2 h, for mice, rats, and dogs, respectively [110]. After intravenous administration, systemic clearance values of 0.68, 0.44, and 0.45 L/h/kg; half-life values of ~5.5, 8.7, and 9.3 h; volume of distribution values of 5.3, 4.7, and 5.1 L/kg, were observed for mice, rats, and dogs, respectively [110]. In vitro assessment of metabolic stability of IBX utilizing rodent, dog, and human liver microsomes gave clearance rates of ≤11, ≤48, and 34 µL/min/mg, respectively; corresponding to clearance rates of <40, ≤69, and 38 µL/min/kg, respectively, when scaled for in vivo intrinsic clearance [101]. Due to the observed long half-life and moderate liver clearance, a once-daily dosage was suggested for clinical trials [101]. For all species, declines in plasma concentrations were linear [110].

In vitro solubility of IBX is inversely correlated with pH; there was good solubility in simulated gastric fluid (SGF; >5.2 mg/mL) and fed-state intestinal fluid (FeSSIF; >3.0 mg/mL), but IBX was only slightly soluble in fasted-state simulated intestinal fluid (FaSSIF). The citrate formulation of IBX provided substantial improvement in solubility, increasing solubility in SGF and FeSSIF to >20 mg/mL and in FaSSIF to >4.2 mg/mL after 24 h [110]. IBX is a lipophilic compound; as such, observed protein binding was very high, similar to the echinocandins [101]. In vitro protein binding and blood-to-plasma ratio in mouse, rat, dog, and human cells gave very high protein binding, ranging from 99.8 to 99.5% [110]. Investigation of tissue distribution in the murine model of invasive candidiasis showed high volume distribution to the kidneys with area under the plasma concentration—time of zero to infinity (AUC_0–∞_) and maximum concentration (C_max_) values 20- to 25-fold greater than those for plasma [110]. Distribution volume is inversely proportional to affinity binding of proteins. The high distribution volumes observed for IBX indicate that while protein binding is high, it is most likely of low affinity; allowing better tissue distribution [110]. Furthermore, IBX is bound mainly to plasma proteins with a blood-to-plasma ratio ranging from 0.5 to 0.7; not erythrocytes, giving IBX excellent properties for treating invasive disease [110]. Utilization of radioactive IBX ([14C]SCY-078), orally or parentally, in albino and pigmented rats, showed extensive distribution to tissues involved in invasive fungal disease, including kidney, lung, liver, spleen, bone marrow, muscle, vaginal tissue, and skin [118]. Tissue-to-blood AUC ratios after oral administration were several times higher in tissue relative to blood; with 54-fold higher concentrations in spleen; 50-fold higher in liver; 31-fold higher in lung; 25-fold higher in bone marrow; 20-fold higher in kidney; 12-fold higher in non-pigmented skin; 18-fold higher in pigmented skin; 9-fold higher in vaginal tissue; 4-fold higher in skeletal muscle [118]. There was limited to no distribution to central nervous system tissues (brain and spinal cord); limited distribution to adipose tissues; variable distribution to the eye (none to the lens, but very well distributed to the uvea) [118]. IBX elimination was shown to be mainly via feces and bile (∼90%); a very small proportion via urine (∼1.5%) [118], probably due to high protein binding [110].

Investigation of cytochrome P450 (CYP) inhibition of IBX showed that it is a substrate for CYP3A4 and an inhibitor of CYP2C8; but has very little effect on other CYP isoforms (IC_50_ values >25μM for 1A2, 2B6, 2C9, 2C19, 2D6) (Wring SA, Park SM, unpublished data) [109,119]. A phase 1, open-label, 2-period crossover study, using a rosiglitazone, a sensitive substrate of CYP2C8 metabolism demonstrated that co-administration of IBX with rosiglitazone did not affect the maximum concentration values for rosiglitazone indicating that there is limited to no inhibition of CYP2C8 [109]. In another phase 1 study, the potential drug-drug interaction between IBX and tacrolimus, a substrate of CYP3A4 as well as a potent immunosuppressive drug used to prevent transplant rejection [120], was assessed [110]. The resultant PK values (AUC_0-∞_: 1.42-fold and C_max_: 1.03-fold) for IBX with tacrolimus or alone were similar, indicating that there was very little interaction between IBX and tacrolimus at therapeutic levels of IBX [110]. However, phase 1 studies using ketoconazole (strong CYP3A inhibitor) and diltiazem (moderate CYP3A4 inhibitor), found moderate to severe effects on IBX (AUC_0–∞_: 5.7-fold, C_max_: 2.5-fold) for ketoconazole and for moderate effects for diltiazem (AUC_0–∞_: 2.5-fold, C_max_: 2.2-fold) [119]. Taken together, these phase 1 studies indicate that IBX has limited potential for interaction with drugs metabolized by cytochrome P450; however, a dose adjustment may be necessary for potent CYP3A4 inhibitors [109,110,119]. 

## 8. Indications and Usage

Most clinical trials have focused on the oral formulation of ibrexafungerp [87]. The use of ibrexafungerp for the treatment for vulvovaginal candidiasis (VVC) and prevention of recurrence of VVC was investigated in six studies that include efficacy (Table 1) [113,114,115,116]. These studies have demonstrated a favourable safety and tolerability profile, as well as high efficacy in the context of VVC [113,114,115], which led to acceptance of a new drug application (NDA), by the US Food and Drug Administration (FDA), for the treatment of VVC using ibrexafungerp [121]. Furthermore, Qualified Infectious Disease Product (QIDP) and Fast Track designations were granted by the FDA for the treatment of VVC and prevention of recurrent VVC with ibrexafungerp [121]. Results from completed clinical trials or preliminary data from ongoing trials have shown inbrexafungerp to be effective for treatment of invasive candidiasis including *C. auris* [108,112]; for use as salvage therapy for refractory fungal infections [117,122]. Treatment of invasive pulmonary aspergillosis as combination therapy with azoles was found to be effective in in vitro [30] and in vivo animal models [106]; however, clinical trials are ongoing [117]. Apart from candidiasis and aspergillosis, IBX is being assessed in the ongoing phase 3 FURI trial, for treatment of other fungal infections such as coccidioidomycosis, histoplasmosis, and blastomycosis [117].

## 9. Conclusions

The rising incidence of IFIs, development of antifungal resistance and emergence of multi-drug resistant species, such as *C. auris,* are of public health concern. Thus, the development and availability of new drugs, such as ibrexafungerp that have activity against the most prevalent fungal pathogens, *Candida* and *Aspergillus*, including biofilm-forming strains, azole- and echinocandin-resistant strains provide alternative treatment options, where previously there were none or few options. The good oral bioavailability and once-daily dosing of ibrexafungerp will reduce the burden of IV administration, unnecessarily prolonged hospitalization, and complex dosing schedules, thereby increasing adherence and the likelihood of treatment success. Furthermore, IBX has favourable characteristics such as very low toxicities, enhanced activity at low pH for better activity in tissues and abscesses [104]; high tissue penetration for invasive disease; low risk of drug–drug interactions that will allow treatment combinations and treatment of patients with multiple comorbidities [14]. IBX is indicated for vulvovaginal candidiasis, invasive candidiasis, invasive aspergillosis, pneumocystosis and some refractory invasive fungal infections [122]. The FDA has granted QIDP and fast track designations for oral and IV formulations of ibrexafungerp for the treatment of invasive candidiasis, vulvovaginal candidiasis and invasive aspergillosis; IBX was also given orphan drug designation for invasive candidiasis and invasive aspergillosis [122]. The FDA has set 1 June 2021 as the target date of action under the prescription drug user fee Act (Figure 2) [14,122]; resulting in a commercial launch in the United States for treatment of VVC being planned in 2021 [123]. The high cost of new antifungals limits their access, especially in low- and middle-income countries (LMICs) where the fungal disease burden is high but the perceived commercial market is small, limiting manufacturers’ interest in additional regulatory approvals [124]. For example, echinocandins were registered in South Africa only 3–10 years after their approval and registration in the United States [4,125,126,127]. Ibrexafungerp patent applications are underway for 10 years of U.S. regulatory exclusivity plus composition-of-matter patent up to 2035, with additional applications pending, for a total of ~15 years of exclusivity in the U.S [123]. This will further delay access of this drug to most LMIC countries; hence early and efficient partnerships among pharmaceutical companies, governments and international organizations are necessary to promote global access of this novel medication.

## Figures and Tables

**Figure 1 jof-07-00163-f001:**
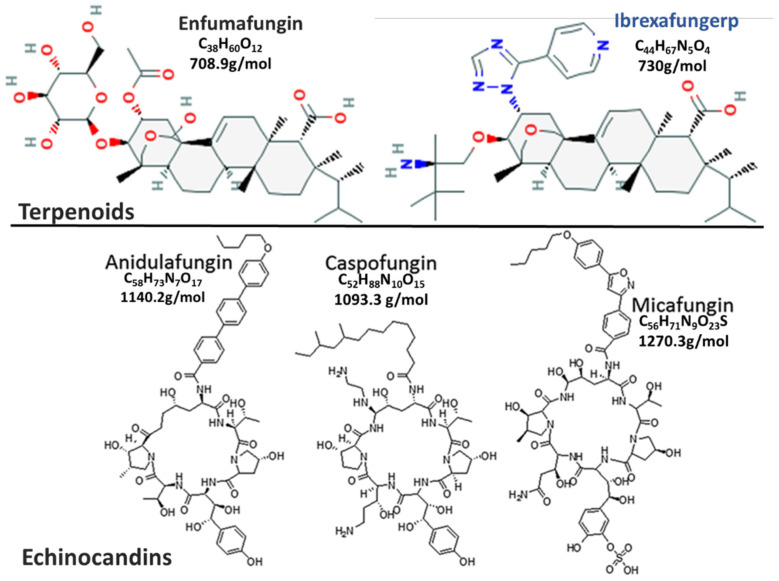
This is a figure comparing Fungerp and Echinocandin chemical structures (modified from [10,11]).

**Figure 2 jof-07-00163-f002:**
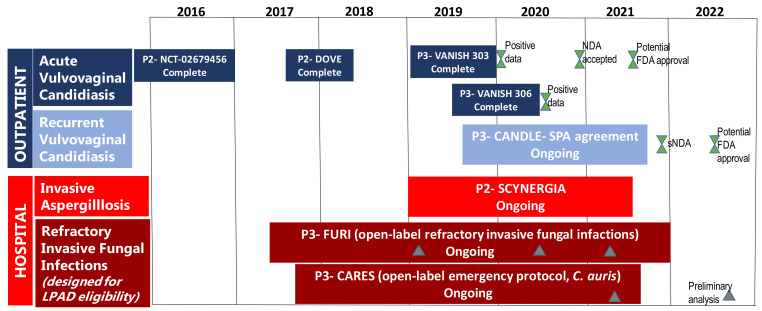
This is a figure of the Ibrexafungerp clinical trials and milestones (modified from [14]). Abbreviations: New Drug Application (NDA), supplemental New Drug Application (sNDA), Limited Population Pathway for Antibacterial and Antifungal Drugs (LPAD), Special Protocol Assessment (SPA), Invasive Fungal Infections (IFI).

**Table 1 jof-07-00163-t001:** This is a table showing the details of current clinical trials involving ibrexafungerp.

Phase	NCT Number	Acronym	Title	Conditions	Drugs	Outcome Measures	Age (yrs)	#	Start Date	End Date
Phase 1	NCT04307082	ADME	ADME Study of [14C]-Ibrexafungerp in Healthy Male Subjects	Fungal Infection	[14C]-Ibrexafungerp (IBX)	Mass balance|Routes & rates of elimination of [14C]-IBX |Number of subjects with treatment-emergent adverse events	30–65	6	5 December 2019	30 June 2020
Phase 1	NCT04092751		Study to Evaluate the Effect of SCY-078 (Ibrexafungerp) on the PK of Pravastatin in Healthy Subjects	Pharmacokinetics	PRA| SCY-078 plus PRA	Pharmacokinetics of PRA + SCY-078: AUC, Cmax, Tmax, Half-life |Safety & tolerability of the oral combination PRA + SCY-078	18–55	28	22 November 2019	20 December 2019
Phase 1	NCT04092725		Study to Evaluate the Effect of SCY-078 on the PK of Dabigatran in Healthy Subjects	Pharmacokinetics	DAB|SCY-078 plus DAB	Pharmacokinetics of DAB + SCY-078: AUC, Cmax, Tmax, Half-life |Safety & tolerability of the oral combination DAB + SCY-078	18–55	36	9 September 2019	3 January 2020
Phase 2	NCT02244606		Oral Ibrexafungerp (SCY-078) vs Standard-of-Care Following IV Echinocandin in the Treatment of Invasive Candidiasis	Mycoses, Candidiasis, Invasive, Candidemia	SCY-078|Fluconazole| Micafungin	Safety & tolerability, assessed by adverse events, clinical laboratory results, physical examination findings, ECG results, & vital sign measurements|Dose of SCY-078 that achieves the target exposure (AUC)|Global response| Clinical response| Microbiological response|Relapse	18–80	27	1 September 2014	August 2016
Phase 2	NCT02679456		Safety and Efficacy of Oral Ibrexafungerp (SCY-078) vs. Oral Fluconazole in Subjects With Vulvovaginal Candidiasis	Vulvovaginal Candidiasis	SCY-078|Fluconazole	% of subjects achieving therapeutic cure at TOC visit (Day 24 +/-3)|% of subjects with recurrence of VVC during the observation period	18–65	96	1 November 2015	5 August 2016
Phase 2	NCT03253094	DOVE	An Active-Controlled, Dose-Finding Study of Oral IBX vs. Oral Fluconazole in Subjects With Acute Vulvovaginal Candidiasis	*Candida* Vulvovaginitis	Fluconazole|SCY-078	Clinical cure (complete resolution of signs & symptoms)|Co-occurrence of clinical & mycological cure	18–100	186	1 August 2017	4 May 2018
Phase 2	NCT03672292	SCYNERGIA	Study to Evaluate the Safety and Efficacy of the Coadministration of Ibrexafungerp (SCY-078) With Voriconazole in Patients With Invasive Pulmonary Aspergillosis	Invasive Pulmonary Aspergillosis	SCY-078|Voriconazole| Oral Placebo Tablets	Adverse events; discontinuation due to AE; death|Composite clinical, radiological & mycological response (global response)| Death| Change in serum GMI|Study & comparator plasma concentrations	≥18	60	22 January 2019	7 June 2021
Phase 3	NCT03987620	Vanish 306	Efficacy and Safety of Oral Ibrexafungerp (SCY-078) vs. Placebo in Subjects With Acute Vulvovaginal Candidiasis	*Candida* Vulvovaginitis	Ibrexafungerp|Placebo	Clinical cure (complete resolution of signs & symptoms)|Mycological eradication (negative culture for growth of yeast)|Clinical cure & mycological eradication (responder outcome)|Complete resolution of signs 7 symptoms at follow-up|Safety & tolerability of IBX	≥12	366	7 June 2019	29 April 2020
Phase 3	NCT03734991	Vanish 303	Efficacy and Safety of Oral Ibrexafungerp (SCY-078) vs. Placebo in Subjects With Acute Vulvovaginal Candidiasis (VANISH 303)	*Candida* Vulvovaginitis	Ibrexafungerp|Placebo	Clinical cure (complete resolution of signs & symptoms) | Mycological eradication (negative culture for yeast growth) |Clinical cure & mycological eradication (responder outcome) |Complete resolution of signs & symptoms at follow-up| subjects with treatment-related adverse events	≥12	376	4 January 2019	4 September 2019
Phase 3	NCT03059992	FURI	Study to Evaluate the Efficacy and Safety of Ibrexafungerp in Patients With Fungal Diseases That Are Refractory to or Intolerant of Standard Antifungal Treatment	Candidiasis (Invasive, Mucocutaneous, Recurrent Vulvovaginal)| Coccidioido- mycosis| Histoplasmosis| Blastomycosis |Aspergillosis (Chronic & Invasive Pulmonary, Allergic Bronchopulmonary |Other Emerging Fungi	Ibrexafungerp	Assessment of Global Response|Assessment of Recurrence of Baseline Fungal Infection|Assessment of survival	≥12	200	1 April 2017	5 December 2021
Phase 3	NCT04029116	CANDLE	Phase 3 Study of Oral Ibrexafungerp (SCY-078) Vs. Placebo in Subjects With Recurrent Vulvovaginal Candidiasis (VVC)	Recurrent Vulvovaginal Candidiasis	Fluconazole Tablet| IBX| Placebo oral tablet	Clinical Success|The percentage of subjects with no Mycologically Proven Recurrence|Safety & tolerability	≥12	320	23 September 2019	September, 2021
Phase 3	NCT03363841	CARES	Open-Label Study to Evaluate the Efficacy and Safety of Oral Ibrexafungerp (SCY-078) in Patients With Candidiasis Caused by *Candida* Auris (CARES)	Candidiasis, Invasive Candidemia	SCY-078	Efficacy as measured by % of subjects with global success at end of treatment|Participants with treatment-related Adverse Events |Participants with Discontinuations due to Adverse Events |Recurrence of Baseline Fungal Infection| Survival	≥18	30	15 November 2017	15 May 2021

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
