# Peer review of "Ibrexafungerp: A First-in-Class Oral Triterpenoid Glucan Synthase Inhibitor"

_jof, 2021, doi:10.3390/jof7030163_

Round 1

Reviewer 1 Report

Nicely and clearly written review.  Very comprehensive and relevant.

Line 29:  Consider revision of this sentence for clarity:

From: …component of most fungi, have the potential to be potent with broad-spectrum activity

To:  …component of most fungi, have the potential to exhibit potent broad-spectrum of activity

Line 32: I would recommend removing the month in this sentence (  …..synthase inhibitors approved for use in January 2001 [4] …..) since it likely happened at a different timepoints in different regions and it does not add much.

Line 51: Recommend consistency in writing β-1,3-D-glucan vs. β-(1,3)-D-glucan

Line 101: Recommend adding examples to better illustrate the message of this sentence: Several fungal pathogens (e.g., XXXXXXXXXXXX, XXXXXXXXX, XXXXXXX) are gaining importance, especially in middle–income coun-101 tries such as South Africa, India, Brazil and Colombia.

Paragraph lines 118-123:  For completeness, recommend the authors to consider mentioning that Pneumocystis infections have also become prevalent in some non-HIV populations and mention some examples.

Lines 144-145:  Verify the sentence ( In vitro analysis of ibrexafungerp showed potent activity against a broad spectrum of >1300 Candida species [41, 78, 82-89].) and consider if rather than >1300 Candida species it should say Candida isolates or Candida strains.

Section 4, Starting in Line 204: High tissue penetration is mentioned by the authors among the key attributes of IBX, the authors should consider briefly including in vivo (mice) published data illustrating how this attribute may translate into efficacy for localized infections (e.g. intra-abdominal)

Lines 208-210:  verify the accuracy of this sentence confirming if parental use of IBX was included in the referenced study: IBX given both orally and parentally showed activity against C. albicans, C. glabrata, and C. parapsilosis in a neutropenic murine model of 209 disseminated candidiasis, [98]

Table 1: Correct typo for head of 6th column: “Drug$”  and the typo of End Date of SCYNERGIA study (from 2012 to 2021)

Section 7: Since efficacy rates for the study in invasive candidiasis (lines 373-375) are already mentioned in section 5 (lines 248-248) it may not be needed to repeat here.

Section 7: This section could be further summarized and organized by the main potential indications starting each paragraph with the potential indication for ease of readers focus and understanding (i.e., treatment of VVC and prevention of recurrence of VVC | salvage therapy for refractory fungal infections | invasive candidiasis including C.auris | invasive pulmonary aspergillosis (as combination therapy)

Line 390:  Consider “alternative” rather than “alternate”

Lines 413-15:  This message is not clear, how the author suggest that generic manufactures involvement will expedite the access of this drug (or any other) to LMIC countries prior to expiration of patent exclusivity? Also, I could not see in the reference (121) that this idea is supported this publication. This is an important and complex topic and the authors could consider making their statement more broadly stating the need for early and efficient partnerships among pharmaceutical companies, government and organizations promoting global access to ensure novel medications intended to address unmet medical needs are rapidly made available in LMIC countries.

Reviewer 2 Report

Dear Authors,

The manuscript ID: jof-1109040 entitled “Ibrexafungerp: A First-in-Class Oral Triterpenoid Glucan Synthase Inhibitor” written by Sabelle Jallow and Nelesh P. Govender is devoted to a new antifungal drug.

The purpose of this review is very interesting. Fungal infections are a major contributor to infectious disease-related deaths across the globe. In turn, the antifungal resistance continues to grow and evolve and complicate patient management. Currently, there are only three major classes of drugs approved for the treatment of invasive mycoses, and the efficacy of these agents is compromised by the development of drug resistance in pathogen populations. Notably, the emergence of additional drug-resistant species, such as Candida auris and Candida glabrata, further threatens the limited armamentarium of antifungals available to treat these serious infections. Thus, a need exists for novel antifungal agents that demonstrate high levels of activity against these microorganisms.

According to me, Authors prepared a good manuscript. The knowledge of the new drug –

ibrexafungerp, was extensively documented. Moreover, numerous current references were used. The whole review is appropriately organized and described.

I have only a small suggestion in order to improve paper:

  • μg/mL or μg/ml – please harmonize throughout the text;
  • 125 - 2 μg/mL or 0.125-2 μg/mL – please harmonize throughout the text spaces or without spaces;
  • Line 146: IBX „generally” was „generally” – please correct;
  • The text and literature should be prepared in accordance with the instructions for authors.

I think, this review is valuable and may be accepted for the publication in “Journal of Fungi”.

With highest regards,
